# Comparison of participant-collected nasal and staff-collected oropharyngeal specimens for human ribonuclease P detection with RT-PCR during a community-based study

Mitchell T. Arnold[ID][1][☯]*, Jonathan L. Temte[1][☯], Shari K. Barlow[1][‡], Cristalyne J. Bell[1][‡], Maureen D. Goss[1][‡], Emily G. Temte[1][‡], Mary M. Checovich[1][‡], Erik Reisdorf[2][‡], Samantha Scott[2][‡], Kyley Guenther[2][‡], Mary Wedig[2][‡], Peter Shult[2][‡], Amra Uzicanin[3][☯]

1 Department of Family Medicine and Community Health, University of Wisconsin-Madison School of Medicine and Public Health, Madison, WI, United States of America, 2 Wisconsin State Laboratory of Hygiene, University of Wisconsin-Madison, Madison, WI, United States of America, 3 US Centers for Disease Control and Prevention, Atlanta, GA, United States of America

☯ These authors contributed equally to this work.
‡ These authors also contributed equally to this work.
* mtarnold@wisc.edu

**Data Availability Statement:** All relevant data are within the manuscript and its Supporting Information files. The ORCHARDS study and the

## Abstract

We analyzed 4,352 participant- and staff-collected respiratory specimens from 2,796 subjects in the Oregon Child Absenteeism due to Respiratory Disease Study. Trained staff collected oropharyngeal specimens from school-aged children with acute respiratory illness while household participants of all ages collected their own midturbinate nasal specimens in year one and anterior nasal specimens in year two. Human ribonuclease P levels were measured using RT-PCR for all staff- and participant-collected specimens to determine adequacy, defined as Cycle threshold less than 38. Overall, staff- and participant-collected specimens were 99.9% and 96.4% adequate, respectively. Participant-collected midturbinate specimens were 95.2% adequate in year one, increasing to 97.2% in year two with anterior nasal collection. The mean human ribonuclease P Cycle threshold for participant-collected specimens was 31.18 in year one and 28.48 in year two. The results from this study suggest that community-based participant collection of respiratory specimens is comparable to staff-collected oropharyngeal specimens, is feasible, and may be optimal with anterior nasal collection.

## Introduction

Influenza remains one of the leading causes of hospitalizations and mortality in the United States [1, 2]. Early laboratory testing and diagnosis of influenza can reduce influenza-related morbidity and mortality by allowing clinicians to quickly implement appropriate isolation precautions and start antiviral therapy [3, 4]. Laboratory confirmation is also essential for the surveillance of circulating influenza viruses during seasonal and pandemic outbreaks, which is

Household Substudy are ongoing and will continue through at least the 2020/2021 influenza season. All data from this extensive study will be shared upon study completion. For this resubmission, we are making available all data that were used in the analysis, following removal of any personal identifiers. The data have been included in the Supplemental Information (S4 File) as a spreadsheet with all record ID numbers, ages, RT-PCR results, and human RP Ct values for each participant for whom collection was completed. We have also compiled two other files (S5 File and S6 File) that include all calculations and a summary of the analyses.

**Funding:** The research reported in this manuscript was supported by the Centers for Disease Control and Prevention through Cooperative Agreement U01 CK000542-01. Its contents are solely the responsibility of the authors and do not necessarily represent the official views of the Centers for Disease Control and Prevention or the Department of Health and Human Services.

**Competing interests:** I have read the journal's policy and the authors of this manuscript have the following competing interest: Dr. Jonathan L. Temte received in-kind research support from Quidel Corporation for the ORCHARDS study. Quidel Corporation did not direct or exert any influence over this manuscript. This does not alter our adherence to all PLOS ONE policies on sharing data and materials.

important for monitoring disease activity, detecting genetic and antigenic changes, and maintaining effective vaccinations, treatments, and public responses to influenza [5].

Respiratory virus surveillance has traditionally relied on medically trained staff and clinicians collecting nasopharyngeal aspirate (NPA) or nasopharyngeal (NP) specimens from patients who present to medical facilities with signs and/or symptoms of acute respiratory infection (ARI). As more sensitive molecular methods have become standardized, less invasive collection techniques, such as oropharyngeal (OP) collection, nasal collection, or a combination of the two have yielded comparable results to both NP and NPA collection [6–13].

Studies have shown that participant collection (defined as collection by oneself, a partner, or a parent) of nasal specimens produces comparable results to collection by trained clinical, laboratory or research staff while reducing collection costs and increasing patient comfort and preference [14–23]. These results have been shown with both midturbinate [15–18] and anterior nasal collection techniques [19–23]. However, few studies have compared staff and participant collection of respiratory specimens in community-based designs with large sample sizes [21–25], and no studies compare participant collection of midturbinate and anterior nasal specimens within the same general population.

The purpose of this study was to use RT-PCR testing to determine the adequacy of participant-collected nasal specimens in a large, community-based sample. A specimen was determined adequate if the RT-PCR cycle threshold (Ct) for the human ribonuclease P (RP) gene was less than 38. We also compared the adequacy of participant collection with two different techniques: midturbinate nasal collection and anterior nasal collection.

## Methods

The Oregon Child Absenteeism due to Respiratory Disease Study (ORCHARDS) is a longitudinal, community-based assessment of the relationship between school absenteeism and influenza in the Oregon School District (Oregon, Wisconsin). This human subject research study was approved by the University of Wisconsin Health Sciences Institutional Review Board Protocol # 2013–1357. Written consent/assent/parental consent was obtained for all human subjects. The protocol is provided at https://dx.doi.org/10.17504/protocols.io.bhsrj6d6.

Parents of children exhibiting respiratory symptoms voluntarily called the ORCHARDS study telephone hotline and were screened by phone to determine eligibility. Students were eligible if they met criteria for ARI. To meet criteria for an ARI, the child must have two or more respiratory symptoms (fever, cough, runny nose, nasal congestion, sore throat, and/or sneezing), a Jackson score of 2 or greater [26], and symptom onset within seven days. Study staff then conducted home visits within seven days of symptom onset for children with ARI. The average time between self-reported symptom onset and home visit time was 2.25 days (SD = 1.42).

During the home visit, staff collected data using a standard study questionnaire (S1 File). Staff then collected an anterior nasal specimen using a Puritan® Sterile Foam Tipped Applicator and an OP specimen, or NP (1.5% of all samples) if OP was not tolerated, using a Copan FLOQSwabs™ flocked swab. Since 98.5% of all staff-collected samples were OP, they will be referred to as OP specimens throughout this paper. The OP swab was immediately placed into Remel Microtest™ M4-RT Viral Transport Medium (VTM). Following rapid influenza diagnostic testing using the Quidel Sofia® Influenza A+B Fluorescent Immunoassay, the nasal swab was placed in the same VTM containing the OP swab before being stored at 2˚C–8˚C. A courier then delivered the VTM to the Wisconsin State Laboratory of Hygiene (WSLH). The average time between staff collection and the time of RT-PCR testing was 1.93 days (SD = 1.08).

During the screening process, families of children who participated in ORCHARDS were invited to participate in a household influenza transmission substudy, which included participant collection of nasal specimens. The protocol for the Household Substudy is provided at dx. doi.org/10.17504/protocols.io.bhssj6ee.

At the home visit, the visiting staff member instructed the interested parents on how to collect a respiratory specimen using a Copan FLOQSwabs™ flocked midturbinate swab (Year 1) or an anterior nasal Puritan® Sterile Foam Tipped Applicator (Year 2). For midturbinate collection, parents were instructed to insert the midturbinate swab inside the nostril to a depth of 2–4 cm and rotate against the walls of the nostril three times. For anterior nasal collection, they were instructed to insert the nasal swab inside the nostril to a depth of 1–2 cm and rotate against the walls of the nostril three times. The differences in size and shape of these swabs are shown in Fig 1.

The staff then left the families with a Household Substudy Kit (Fig 2) for each family member that included an ORCHARDS Household Substudy Questionnaire (S2 File), two nasal/ midturbinate swabs, two VTMs, a biohazard bag, and instructions (S3 File) for collection. Parents were instructed to collect specimens from all household members without staff observation the day of the home visit (Day 0) and again seven days later (Day 7), immediately placing each swab into a color coded (green for Day 0 and red for Day 7) VTM tube and keeping all specimens refrigerated. Staff then retrieved and delivered the participant-collected specimens to the WSLH on or shortly after Day 7.

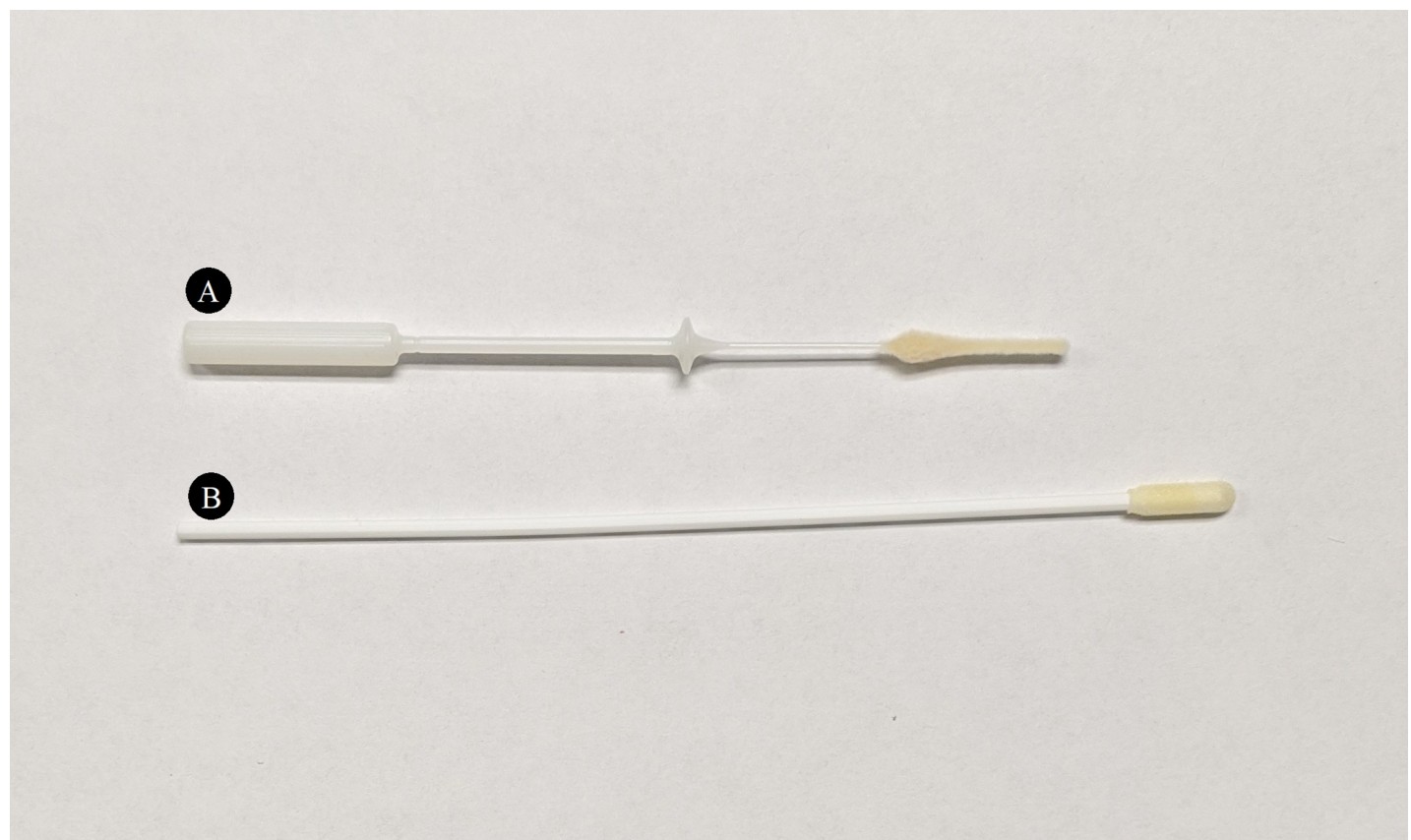

**Fig 1. Comparison of the midturbinate swab and the anterior nasal swab.** Photograph of the (A) Copan FLOQSwabs™ flocked midturbinate swab and the (B) anterior nasal Puritan® Sterile Foam Tipped Applicator.

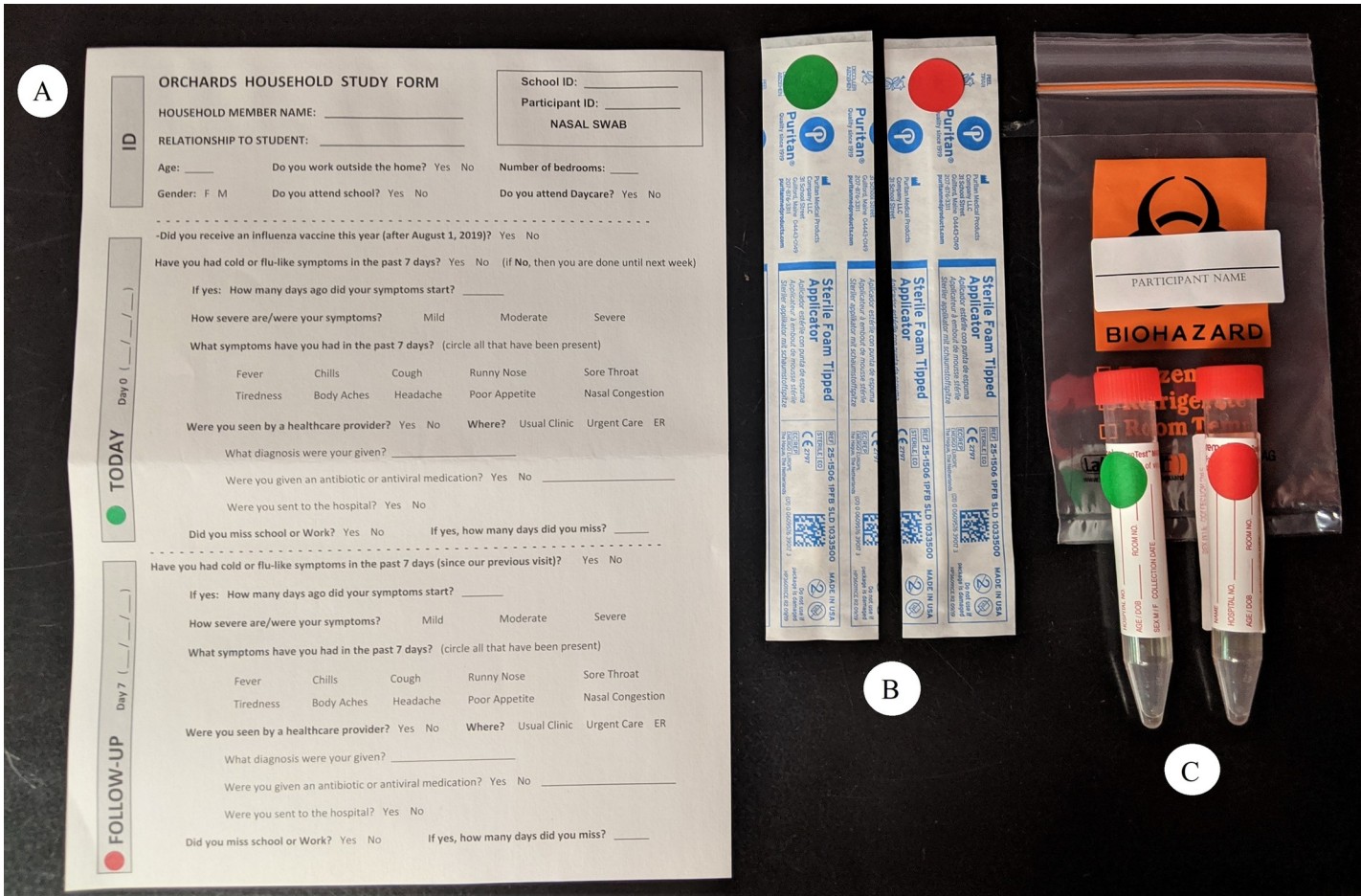

**Fig 2. Deconstructed Household Substudy Kit.** Each household participant receives a (A) Household Substudy questionnaire, (B) two midturbinate or anterior nasal swabs color coded to represent Day 0 and Day 7 (anterior nasal shown), (C) two VTMs color coded to represent Day 0 and Day 7, and a biohazard bag to place the collected specimens in.

All specimens were tested at the WSLH for influenza A and B viruses and the human RP gene using the in vitro diagnostic Food and Drug Administration-approved Centers for Disease Control and Prevention Human Influenza Virus Real-time RT-PCR Diagnostic Panel (Cat. No. FluiVD03) [27]. Detection of the human RP gene indicates that adequate isolation of nucleic acid resulted from the extraction of the clinical specimen. The RP Ct values for each specimen were recorded and determined adequate if Ct<38, following recommendations by the CDC RT-PCR protocol [27]. A lower RP Ct value would indicate a higher amount of collected human RP and thus a better specimen. Samples with Ct>38 or no observable human RP were classified as inconclusive samples. If a participant returned a VTM without a swab or the VTM had a leak, the specimen was not tested. If the sample was positive for influenza RNA but negative for human RP, it was omitted from the analysis.

Confidence intervals for the percent adequacy and mean RP Ct value of the specimens were then calculated for each group and for each year of the study. Participant-collected specimens from school-aged children were analyzed separately to account for age differences between the participant- and staff-collected groups. To analyze sample stability, separate confidence intervals were also calculated to identify differences in adequacy and mean RP Ct value between participant-collected specimens on Day 0 and Day 7. All confidence intervals used a

confidence level of 95% and a Student's t distribution. Confidence intervals for adequacy used a one sample, dichotomous outcome method for sample proportion while confidence intervals for Ct values used a one sample, continuous outcome method for the sample mean. Mean human RP Ct value was calculated using only the adequate samples with Ct<38.

## Results

During the two-year study, 4,352 swabs were collected from 2,796 new and repeat participants. All participants lived or attended school in the Oregon School District located in south-central Wisconsin. There were slightly more male (54%) participants than female participants. All participants within the staff-collected groups were school-aged children between the ages of 4 and 18 years (median = 9 years, mean = 9.8 years, SD = 3.4 years). There were no age restrictions for the participant-collected groups, resulting in 4% of the participants below the age of 4 years, 60% between ages 4 and 18 years, and 31% between ages 25 and 49 years.

In year one, trained staff collected 376 OP specimens while 740 household participants collected 1,454 midturbinate specimens (Fig 3). Of the participant-collected midturbinate specimens, the lab tested 1,424 while 30 were untested for human RP (16 were incomplete/uncollected, 10 were throat collections, 2 were due to leaking VTMs, and 2 returned a Ct value for influenza but not for human RP and were omitted from analysis). In year two, staff collected 424 OP specimens while 1,256 household participants collected 2,155 anterior nasal specimens. Of the participant-collected anterior nasal specimens, the lab tested 2,128 while 27 were untested for human RP (14 were incomplete/uncollected, 11 were due to leaking VTMs, and 2 returned a Ct value for influenza but not for human RP and were omitted from analysis). In total, there were 800 staff-collected OP specimens and 3,552 participant-collected specimens in the analysis of this study.

Over the 2-year period, RT-PCR analysis of human RP for staff collection of 800 specimens yielded just one inadequate sample (adequacy = 99.9%, 95% CI: 99.6%, 100.0%) with a mean RP Ct value of 27.88 (95% CI: 27.70, 28.06). Analysis of the 3,552 participant-collected samples yielded 129 inadequate samples (adequacy = 96.4%, 95% CI: 95.8%, 97.0%) and a mean RP Ct value of 29.55 (95% CI: 29.43, 29.66).

In year one, staff-collected OP specimens yielded 100% adequacy and a mean RP Ct value of 28.58. Participant-collected midturbinate collection yielded 1,355 adequate and 69 inconclusive samples, resulting in adequacies of 95.2% for all ages and 95.3% for school-aged children,

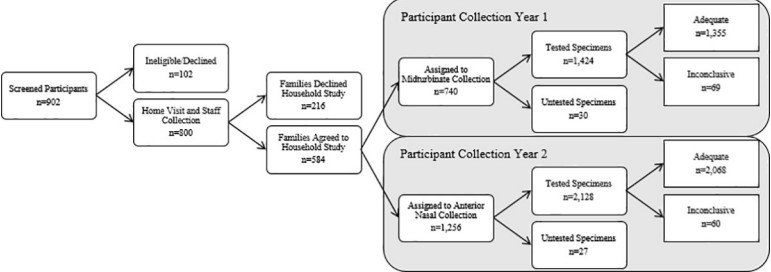

**Fig 3. Study participant and specimen flowchart.** From left to right and top to bottom, this flowchart shows the number of screened participants across both years of the study, the number of participants who declined or were ineligible for the ORCHARDS study, the number who were eligible for the ORCHARDS study and staff collection was conducted, the number of families who declined the Household Substudy, the number of families who agreed to participate in the Household Substudy, the number of participants assigned to midturbinate collection, the number of participants assigned to anterior nasal collection, the number of specimens that were collected and tested by RT-PCR, the number of specimens that were not tested (incomplete collection, leaking VTM, or positive influenza Ct with a negative human RP Ct), the number of adequate specimens, and the number of inconclusive/inadequate specimens.

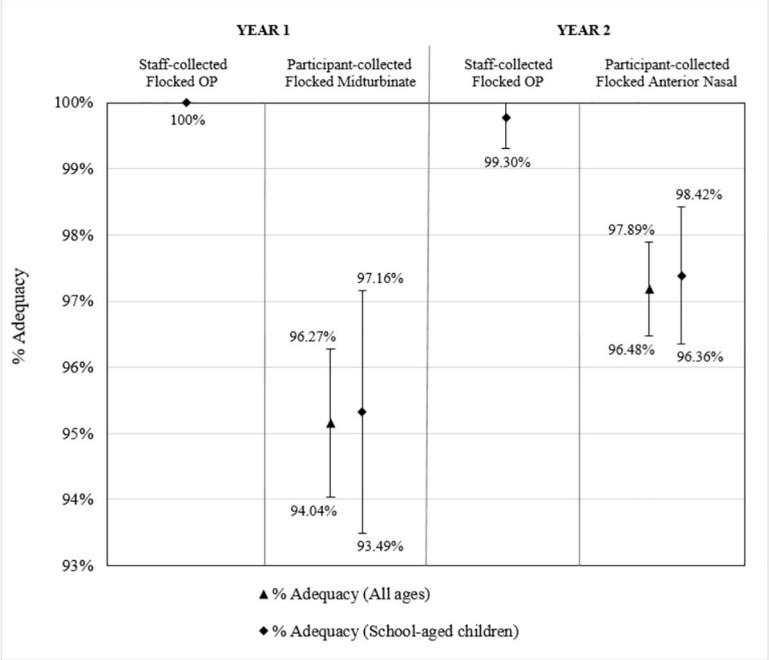

**Fig 4. Adequacies of staff- and participant-collected respiratory specimens.** Bars present the 95% confidence intervals for the percentage of adequate samples (Ct<38) of (from left to right) staff-collected oropharyngeal (OP) specimens from school-aged children in Year 1 using Copan FLOQSwabs™ flocked swabs, participant-collected midturbinate specimens from participants of all ages in Year 1 using Copan FLOQSwabs™ flocked midturbinate swabs, participant-collected midturbinate specimens from school-aged children (ages 4–18) in Year 1 using Copan FLOQSwabs™ flocked midturbinate swabs, staff-collected OP specimens from school-aged children in Year 2 using Copan FLOQSwabs™ flocked swabs, participant-collected anterior nasal specimens from participants of all ages in Year 2 using Puritan Sterile Foam Tipped Applicators, and participant-collected anterior nasal specimens from school-aged children in Year 2 using Puritan Sterile Foam Tipped Applicators.

with mean RP Ct values of 31.18 for all ages and 31.16 for school-aged children. The 95% confidence intervals show that staff OP collection in Year 1 had a significantly higher proportion of adequate samples (Fig 4) and a significantly lower mean RP Ct value (Fig 5) than participant-collected midturbinate samples.

In year two, staff-collected OP samples yielded 1 inadequate sample of 424 samples, resulting in a 99.8% adequacy and a mean RP Ct value of 27.25 (Fig 4). Participant-collected anterior nasal samples resulted in adequacies of 97.2% for all ages and 97.4% for school-aged children, with mean RP Ct values of 28.48 for all ages and 28.16 for school-aged children. The 95% confidence intervals show that staff OP collection in Year 2 had a significantly higher proportion of adequate samples (Fig 4) and a significantly lower mean RP Ct value (Fig 5) than participant-collected anterior nasal samples.

When comparing the two participant collection methods, anterior nasal collection resulted in a significantly higher adequacy (Fig 4) and a significantly lower mean RP Ct value (Fig 5) than midturbinate collection for participants of all ages. Among school-aged children only, anterior nasal collection resulted in a significantly lower mean RP Ct value, but there was no statistically significant difference in the proportion of adequate samples.

Additionally, Day 0 and Day 7 specimens were analyzed to identify if storage in participants' refrigerators for seven days affected specimen stability as measured by adequacy. The average time delay between participant collection on Day 0 and laboratory analysis was 9.89 days (SD = 1.46), and the average time delay between participant collection on Day 7 and

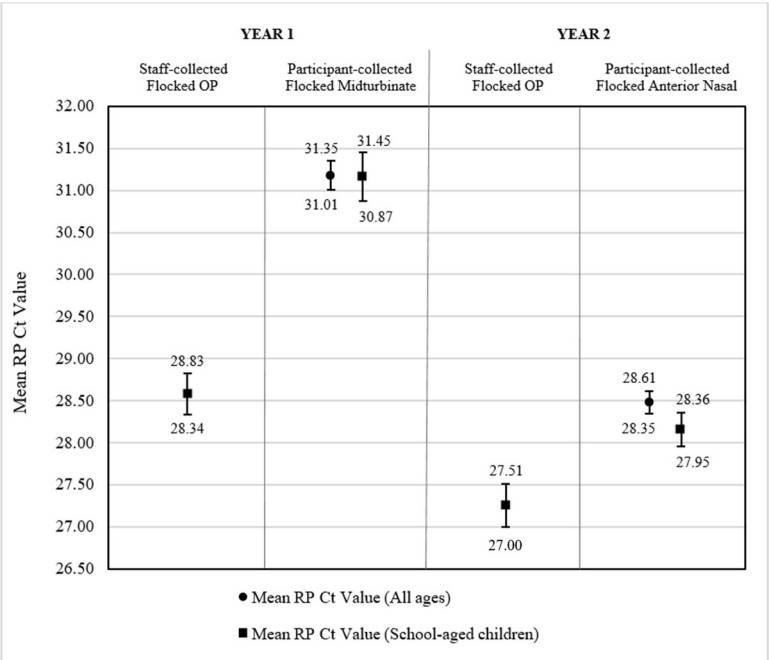

**Fig 5. Mean RP Ct values of staff- and participant-collected respiratory specimens.** Bars present the 95% confidence intervals for the mean human ribonuclease P (RP) cycle threshold (Ct) values of (from left to right) staff-collected oropharyngeal (OP) specimens from school-aged children in Year 1 using Copan FLOQSwabs™ flocked swabs, participant-collected midturbinate specimens from participants of all ages in Year 1 using Copan FLOQSwabs™ flocked midturbinate swabs, participant-collected midturbinate specimens from school-aged children (ages 4–18) in Year 1 using Copan FLOQSwabs™ flocked midturbinate swabs, staff-collected OP specimens from school-aged children in Year 2 using Copan FLOQSwabs™ flocked swabs, participant-collected anterior nasal specimens from participants of all ages in Year 2 using Puritan Sterile Foam Tipped Applicators, and participant-collected anterior nasal specimens from school-aged children in Year 2 using Puritan Sterile Foam Tipped Applicators.

analysis was 2.90 days (SD = 1.44). There were no statistically significant differences in percent adequacy (Fig 6) or mean RP Ct value (Fig 7) when comparing Day 0 and Day 7 specimens for midturbinate collection and anterior nasal collection).

## Discussion

Although staff OP collection had greater performance in terms of adequacy and mean RP Ct value, participant-collected nasal specimens resulted in an overall adequacy of 96.4%. For participant collection from subjects of all ages, anterior nasal specimens were more adequate and had a lower mean RP Ct value than midturbinate specimens. When comparing school-aged children only, anterior nasal specimens had a significantly lower mean RP Ct value than midturbinate specimens, but there was no statistical difference in the proportion of adequate samples. Our secondary findings show that participant-collected respiratory specimens are stable in home refrigerators for up to 7 days. Overall, these results indicate that participant collection of nasal specimens is both feasible and comparable to staff collection of OP specimens, with anterior nasal collection collecting more human RP than midturbinate collection.

These results are consistent with literature findings that show adequate participant collection with both midturbinate and anterior nasal specimens [14–23]. Thompson et al. showed that although RP Ct was higher in participant-collected anterior nasal specimens compared to staff-collected NP specimens, they were 100% adequate for human RP among all 53 subjects [18]. Goyal et al. found that participant-collected anterior nasal samples were 92% adequate

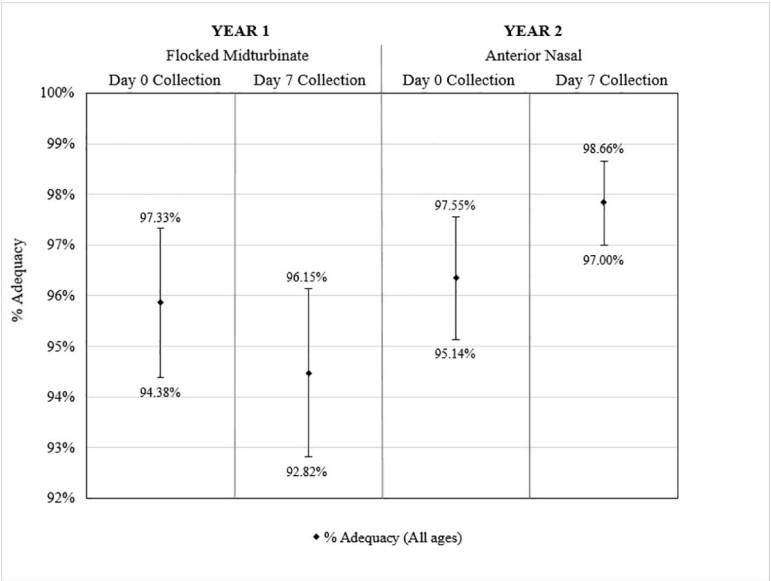

**Fig 6. Adequacies of participant-collected respiratory specimens on Day 0 and Day 7 of the Household Study.** Bars present the 95% confidence intervals for the percentage of adequate samples (Ct<38) of (from left to right) participant-collected midturbinate specimens from participants of all ages in Year 1 on Day 0, participant-collected midturbinate specimens from participants of all ages in Year 1 on Day 7, participant-collected anterior nasal specimens from participants of all ages in Year 2 on Day 0, and participant-collected anterior nasal specimens from participants of all ages in Year 2 on Day 7.

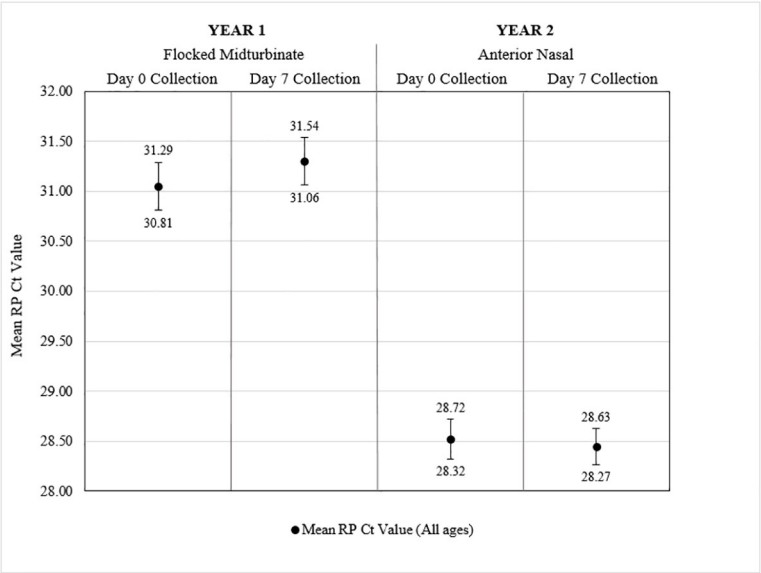

**Fig 7. Mean RP Ct values of participant-collected respiratory specimens on Day 0 and Day 7 of the Household Study.** Bars present the 95% confidence intervals for the mean human ribonuclease P (RP) cycle threshold (Ct) values of (from left to right) participant-collected midturbinate specimens from participants of all ages in Year 1 on Day 0, participant-collected midturbinate specimens from participants of all ages in Year 1 on Day 7, participant-collected anterior nasal specimens from participants of all ages in Year 2 on Day 0, and participant-collected anterior nasal specimens from participants of all ages in Year 2 on Day 7.

among 24 community-based subjects and 99% adequate among 88 medically attended subjects using human RP for a measure of adequacy [24]. Using beta actin as a surrogate for sample adequacy, Smieja et al. showed that participant-collected midturbinate specimens were equivalent to staff-collected NP specimens and Akmatov et al. showed that participant-collected anterior nasal specimens were equivalent to staff-collected anterior nasal specimens [17, 19]. Our study adds a very large sample of 4,352 specimens from 2,786 subjects to strengthen the conclusions that the adequacy of participant-collected nasal specimens is comparable to staff collection.

Rather than using virus detection to measure sample adequacy, this study used human RP as a surrogate because specimens collected by staff originated from symptomatic participants while those in the participant-collected groups were not required to be symptomatic. Accordingly, any comparison of viral detection would be confounded. Human RP, however, is found in epithelial cells throughout the respiratory tract regardless of infection. Since influenza replicates in epithelial cells throughout the respiratory tree [28], human RP is frequently used as a surrogate for viral detection to assess specimen adequacy. The CDC RT-PCR protocol uses human RP as a measure of sample adequacy, regardless of viral detection [27]. Although an NP or NPA specimen may result in higher amounts of epithelial yield [9, 18] multiple studies have found that analysis with advanced molecular methods, such as RT-PCR, results equivalent influenza detection when comparing nasal and NP/NPA specimens [7, 8, 11].

The results in this study also show that utilizing participant collection in a community-based design is a feasible method of increasing participation in research and surveillance. Our research staff was able to collect 800 OP swabs over two years; with participant collection, we were able to obtain an additional 3,552 nasal specimens from family members of children with ARI. This method reduced the costs of additional staff and resulted in the collection of over four times as many respiratory specimens with only slightly lower, yet comparable, adequacy for human RP. This is the largest known sample size for the comparison of participant- and staff-collected specimens in a community-based study. When combined with the conclusions of similar community-based studies, it further validates the feasibility of using participant-collected nasal specimens in research and surveillance [21–23, 29].

Unique to this study, we compared the performance of participant collection with both anterior nasal and midturbinate collection techniques. The change of technique was initiated due to concerns from the WSLH regarding inadequate specimens in year one. Each technique tested participant collection from a different nasal location and with a different type of swab. Anterior nasal collection with the Puritan® Sterile Foam Tipped Applicator resulted in better performance than midturbinate collection with the Copan FLOQSwabs™ flocked midturbinate swab, suggesting that anterior nasal collection may be optimal for participant-collected nasal specimens. It is possible that the greater rigidity of the anterior nasal swab allows participants to push harder along the interior walls of the nostrils, as noted previously [19], compared to the more flexible midturbinate swab [15]. In addition, Emerson et al. found that 94% of participants preferred anterior nasal collection to the deeper midturbinate collection [20]. This increased comfort may have contributed to the increased adequacy seen in our findings. Taking this a step further, one study found that participants preferred collecting their own saliva samples over nasal samples and both had similar viral detection rates [29]. As molecular testing advances, community-based respiratory research should prioritize collection methods that have high preferences among participants to ensure feasibility and compliance.

Finally, we tested the stability of RP in Day 0 and Day 7 specimens and found that there was no difference in specimen adequacy or mean RP Ct when tested after an average of 9.9 and 2.9 days, respectively, following collection. Dare et al. found that maintaining samples at 4°C (39.2°F) for up to 4 days did not affect the presence or amount of influenza and human RNA

[30]. Our study shows that when maintained in participants' home refrigerators at around 4–5˚C, participant-collected nasal samples are 96.4% adequate for human RP for at least 9.9 days between collection and RT-PCR testing. This finding supports the use of at-home participant collection, especially when there may be a time delay between collection and RT-PCR testing or when a series of specimens may be required.

Our results should be considered in the context of at least three limitations. First, according to US census data, the Oregon, Wisconsin community, from which the samples were taken, is more educated and affluent than national averages. Second, participants were recruited from an ongoing study with possible opportunities for repeat training and collection, and consequently, the study population may have been more comfortable with specimen collection than the general population. Certain households may be better or worse at collection than others, which may have led to clustering of specimen quality by household. This was not adjusted for in the analysis. Finally, following rapid influenza testing and prior to RT-PCR testing, the staff-collected nasal specimen was immediately placed in the VTM containing the OP specimen, potentially increasing the quantity of human RP in the staff-collected samples. However, it is likely that removing this step from the protocol would only have decreased the difference in human RP between staff and participant collection, thus strengthening our conclusions.

## Conclusions

Participant collection of nasal swab specimens with minimal training may be sufficient for community-based respiratory virus research/surveillance and may be optimal with less invasive anterior nasal collection. Expanding surveillance programs to include at-home participant collection could be a cost-effective way to increase participation, expand detection to non-medically attended cases, and improve public health responses. At-home collection of respiratory specimens also has the potential to isolate influenza cases, thereby limiting the spread of influenza in hospitals and clinics, preventing transmission to vulnerable patients, and reducing influenza-related morbidity and mortality. Further studies are needed to determine the adequacy of participant nasal collection for influenza detection without any interactions with trained study staff.

## Supporting information

**S1 File. The ORCHARDS home visit questionnaire.** The staff member fills out this questionnaire at the home visit with the help of the ORCHARDS participant and parent(s).
(PDF)

**S2 File. The ORCHARDS Household Substudy questionnaire.** Each Household Substudy participant receives this questionnaire in the Household Substudy Kit and fills it out on Day 0 and Day 7 by him/herself or with the help of a parent.
(PDF)

**S3 File. Household Substudy Kit instructions.** Each Household Study Kit came with instructions for collection that participants could voluntarily use for guidance.
(PDF)

**S4 File. Study data.** A spreadsheet that displays the Record ID #, age, RT-PCR result, and Human RP Ct value for each participant. All identifying factors and specific dates removed.
(XLSX)

**S5 File. Study data with analysis and calculations.** A spreadsheet with all values and calculations shown for the comparison between staff-collected and participant-collected

specimens. These values were used to calculate all demographics, counts, point estimates, and the confidence intervals shown in Figs 4 and 5. All identifying factors and specific dates removed.
(XLSX)

**S6 File. Day 0 and Day 7 analysis and calculations.** A spreadsheet with all values and calculations shown for the comparison between participant-collected specimens on Day 0 and Day 7. These values were used to calculate the confidence intervals shown in Figs 6 and 7. All identifying factors and specific dates removed.
(XLSX)

## Acknowledgments

We gratefully thank Rich Griesser, Tim Davis, Tonya Danz, and Erika Hanson (Virology Laboratory Staff at the Wisconsin State Laboratory of Hygiene) for their assistance with specimen testing throughout the Oregon Child Absenteeism due to Respiratory Disease Study.

## Author Contributions

**Conceptualization:** Mitchell T. Arnold, Jonathan L. Temte, Shari K. Barlow, Mary M. Checovich, Amra Uzicanin.

**Data curation:** Mitchell T. Arnold, Cristalyne J. Bell, Maureen D. Goss, Emily G. Temte, Erik Reisdorf, Samantha Scott, Kyley Guenther, Mary Wedig, Peter Shult.

**Formal analysis:** Mitchell T. Arnold.

**Funding acquisition:** Jonathan L. Temte, Shari K. Barlow, Mary M. Checovich, Amra Uzicanin.

**Investigation:** Mitchell T. Arnold, Jonathan L. Temte, Cristalyne J. Bell, Maureen D. Goss, Emily G. Temte, Erik Reisdorf, Samantha Scott, Kyley Guenther, Mary Wedig, Peter Shult.

**Methodology:** Mitchell T. Arnold, Jonathan L. Temte, Cristalyne J. Bell, Maureen D. Goss, Emily G. Temte, Erik Reisdorf, Samantha Scott, Kyley Guenther, Mary Wedig, Peter Shult.

**Project administration:** Jonathan L. Temte, Shari K. Barlow, Maureen D. Goss, Mary M. Checovich, Amra Uzicanin.

**Resources:** Shari K. Barlow, Emily G. Temte, Mary M. Checovich, Peter Shult.

**Software:** Mitchell T. Arnold.

**Supervision:** Mitchell T. Arnold, Jonathan L. Temte, Shari K. Barlow, Peter Shult, Amra Uzicanin.

**Validation:** Mitchell T. Arnold, Cristalyne J. Bell, Maureen D. Goss, Emily G. Temte, Erik Reisdorf, Samantha Scott, Kyley Guenther, Mary Wedig, Peter Shult.

**Visualization:** Mitchell T. Arnold, Jonathan L. Temte, Amra Uzicanin.

**Writing – original draft:** Mitchell T. Arnold.

**Writing – review & editing:** Mitchell T. Arnold, Jonathan L. Temte, Cristalyne J. Bell, Maureen D. Goss, Emily G. Temte, Mary M. Checovich, Erik Reisdorf, Samantha Scott, Kyley Guenther, Mary Wedig, Peter Shult, Amra Uzicanin.

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
