## [Decision Letter · Decision Letter 0]

19 May 2020

PONE-D-20-10148

Comparison of participant-collected nasal and staff-collected oropharyngeal specimens for human ribonuclease P detection with RT-PCR during a community-based study

PLOS ONE

Dear Mr. Arnold,

Thank you for submitting your manuscript to PLOS ONE. After careful consideration, we feel that it has merit but does not fully meet PLOS ONE’s publication criteria as it currently stands. Therefore, we invite you to submit a revised version of the manuscript that addresses the points raised during the review process.

Please respond to both reviewer comments on a point-by-point basis and amend the manuscript accordingly.

We would appreciate receiving your revised manuscript by Jul 03 2020 11:59PM. To enhance the reproducibility of your results, we recommend that if applicable you deposit your laboratory protocols in protocols.io, where a protocol can be assigned its own identifier (DOI) such that it can be cited independently in the future. For instructions see: http://journals.plos.org/plosone/s/submission-guidelines#loc-laboratory-protocols

We look forward to receiving your revised manuscript.

Kind regards,

Jeffrey Shaman

Academic Editor

PLOS ONE

Journal Requirements:

2. Please include additional information regarding the survey or questionnaire used in the study and ensure that you have provided sufficient details that others could replicate the analyses.

For instance, if you developed a questionnaire as part of this study and it is not under a copyright more restrictive than CC-BY, please include a copy, in both the original language and English, as Supporting Information.

3.  Thank you for including your ethics statement:

"Human Subject Research: University of WI – Health Sciences Institutional Review Board – Protocol # 2013-1357. Written informed consent/assent/parental consent was obtained for all human subjects."

a. Please amend your current ethics statement to confirm that your named institutional review board or ethics committee specifically approved this study.

b. Once you have amended this statement in the Methods section of the manuscript, please add the same text to the “Ethics Statement” field of the submission form (via “Edit Submission”).

'I have read the journal's policy and the authors of this manuscript have the following competing interest: Dr. Jonathan Temte received in-kind research support from Quidel Corporation for the ORCHARDS study. Quidel Corporation did not direct or exert any influence over this manuscript.'

Reviewers' comments:

Reviewer's Responses to Questions

**Comments to the Author**

1. Is the manuscript technically sound, and do the data support the conclusions?

Reviewer #1: Yes

Reviewer #2: Partly

2. Has the statistical analysis been performed appropriately and rigorously? 

Reviewer #1: Yes

Reviewer #2: Yes

3. Have the authors made all data underlying the findings in their manuscript fully available?

Reviewer #1: No

Reviewer #2: Yes

4. Is the manuscript presented in an intelligible fashion and written in standard English?

Reviewer #1: Yes

Reviewer #2: Yes

5. Review Comments to the Author

Reviewer #1: This paper discusses the quality of specimens collected via nasal swabs across multiple axes (participant versus staff collected, delay between specimen collection and testing). Results are very pertinent to influenza as well as other respiratory infection outbreaks.

Overall the study is very clearly described and systematically performed in this paper. There are very minor suggestions:

*Would be helpful to specify the criteria for ARI (line 70)

*Some description regarding what is involved/differences in the anterior vs midturbindate swab collection would be helpful for the non-clinical reader. Consider adding text or a figure if available.

*Would putting the nasal and OP swabs in the same medium conflate results at all? (line 78)

Several pertinent references to previous work are included. Consider citing this paper which shows concordant results in general, but also differs in terms of if the patient self-reports/decides to swab

Goff, Jennifer, et al. "Surveillance of acute respiratory infections using community-submitted symptoms and specimens for molecular diagnostic testing." PLoS currents 7 (2015).

*Some reasoning around the Ct threshold of 38 would be helpful to include

*Paragraph starting at line 125, consider including that differences are statistically significant where appropriate

Note: I am concerned about the data sharing. A broad response was given that all data from the Oregon study will be shared, as well as a comment that "some restrictions may apply". It would be helpful to clarify what the restrictions would be and why. Also, to clarify how long after study completion the data will be made available.

Reviewer #2: This study reports the results of an analysis of specimen quality from self-collected mid-turbinate or nasal swab specimens compared to staff collected oropharyngeal specimens for respiratory virus identification. The manuscript is well written. The methods could be more thoroughly explained, especially with regard to statistical comparisons. I think additional tables/figures would help with the presentation of data and the interpretation of the results. I have additional suggestions by manuscript section, listed below:

Methods

How were confidence intervals calculated? Bootstrap estimates? Were CIs the only statistical means by which differences were tested?

Results

I think some further explanation of the sampling would be helpful or even the study population in general. How many household contacts were swabbed etc? For example: Based on the reported numbers in the results, I assume that there were 376 ARI cases reported in school aged children who were swabbed and then enrolled in the follow up study, it then appears that 712 household members collected two follow up swabs, presumably this includes some households (or individual members) who were only able to collect one specimen. Additionally, the original 376 school aged children are part of the group with self-collected specimens. The same questions would apply to year two. Further there could be clustering of specimen quality by household. So it would be good to have a table or flow chart with the number of school-aged ARI cases that triggered collection, the number of households and household contacts, and the number of complete sets of specimens collected.

Figures

I would separate figures into Ct values and adequacy rather than including both on the same plot. Perhaps make separate panels for each comparison. I also was confused by place Year 1 and Year 2 next to each other in the Figure 1 plot, when the comparison of interest is between

6. PLOS authors have the option to publish the peer review history of their article (what does this mean?). If published, this will include your full peer review and any attached files.

Reviewer #1: No

Reviewer #2: No

---

## [Author Response · Author response to Decision Letter 0]

27 Jul 2020

The responses to the editor and reviewer comments can be found in the "Response to reviewers" document uploaded with this submission. The letter has been copied and pasted here, but it may be easier to read the responses in the provided document.

29 June 2020

Jeffrey Shaman

Academic Editor

PLOS ONE

Manuscript Ref: PONE-D-20-10148

Comparison of participant-collected nasal and staff-collected oropharyngeal specimens for human ribonuclease P detection with RT-PCR during a community-based study

Dear Mr. Shaman:

We write to express our gratitude for the opportunity to revise and resubmit our manuscript, “Comparison of participant-collected nasal and staff-collected oropharyngeal specimens for human ribonuclease P detection with RT-PCR during a community-based study,” for consideration by PLOS ONE. Below we respond to all of the comments and questions proposed by the Editor and both Reviewers.

Editor Comments:

All requirements have been identified and followed, including requirements for file naming.

2. Please include additional information regarding the survey or questionnaire used in the study and ensure that you have provided sufficient details that others could replicate the analyses.

The questionnaires used in the ORCHARDS Study and the Household Substudy were added to the Supplemental Information section as S1 File and S2 File.

3. Thank you for including your ethics statement:

"Human Subject Research: University of WI – Health Sciences Institutional Review Board – Protocol # 2013-1357. Written informed consent/assent/parental consent was obtained for all human subjects."

a. Please amend your current ethics statement to confirm that your named institutional review board or ethics committee specifically approved this study.

b. Once you have amended this statement in the Methods section of the manuscript, please add the same text to the “Ethics Statement” field of the submission form (via “Edit Submission”).

The above text was changed to: “This human subject research study was approved by the University of Wisconsin Health Sciences Institutional Review Board Protocol # 2013-1357. Written consent/assent/parental consent was obtained for all human subjects.” It was added to the Methods section on lines 64-67 and was also added to the “Ethics Statement” field of the resubmission form.

The ORCHARDS study and the Household Substudy are ongoing and will continue through at least the 2020/2021 influenza season. All data from this extensive study will be shared upon study completion. For this resubmission, we are making available all data that were used in the analysis, following removal of any personal identifiers. The data have been included in the Supplemental Information (S4 File) as a spreadsheet with all record ID numbers, ages, collection dates, testing dates, RT-PCR results, and human RP Ct values for each participant for whom collection was completed. We also included the DOIs for the protocols for the ORCHARDS Study and the Household Substudy on lines 67 and 89 of the Methods section as recommended.

'I have read the journal's policy and the authors of this manuscript have the following competing interest: Dr. Jonathan Temte received in-kind research support from Quidel Corporation for the ORCHARDS study. Quidel Corporation did not direct or exert any influence over this manuscript.'

We have updated the Competing Interests section to “I have read the journal's policy and the authors of this manuscript have the following competing interest: Dr. Jonathan Temte received in-kind research support from Quidel Corporation for the ORCHARDS study. Quidel Corporation did not direct or exert any influence over this manuscript. This does not alter our adherence to all PLOS ONE policies on sharing data and materials.” This has been changed on the updated Cover Letter.

Reviewer #1:

1. Would be helpful to specify the criteria for ARI (line 70)

We agree and added the criteria for ARI on lines 70-72.

2. Some description regarding what is involved/differences in the anterior vs midturbinate swab collection would be helpful for the non-clinical reader. Consider adding text or a figure if available.

We agree that more description of these two swabs in the methods section would be helpful. We added a photograph of the two swabs next to each other (Fig 1) as well as more description of the different swabbing methods and depths on lines 91-95. Additionally, we added the written instructions that were in each Household Substudy Kit to the Supplemental Information section as S3 File.

3. Would putting the nasal and OP swabs in the same medium conflate the results at all?

Yes, this is addressed in the limitations on lines 317-321. After a rapid test was conducted on the staff-collected nasal swab, the residual swab was placed in the VTM containing the OP swab. This was done to ensure the best possible specimen for clinical influenza testing and reporting. Although it is not a true one-to-one comparison, this step likely had a positive effect on the amount of human RP in the staff-collected samples. Removing this step would likely have decreased the differences in human RP between the staff- and participant-collected samples, thus strengthening the conclusions of this study.

4. Several pertinent references to previous work are included. Consider citing this paper which shows concordant results in general, but also differs in terms of if the patient self-reports/decides to swab. 

Goff, Jennifer et al. “Surveillance of acute respiratory infections using community-submitted symptoms and specimens for molecular diagnostic testing.” PLoS currents 7 (2015).

The provided study used a cohort of 295 participants that mailed-in test kits when they became symptomatic. It compared participant-collected saliva vs flocked nasal samples. It also included a survey about the feasibility of collection. The major finding is that participants were willing and able to collect their own specimens by both nasal and saliva methods with slight preference for saliva. We find that this reference best fits in our discussion on lines 275-284 and reinforces this statement: “When combined with the conclusions of similar community-based studies, it further validates the feasibility of using participant-collected nasal specimens in research and surveillance.” We also added lines 296-298 and used this reference to recommend the use of participant-preferred collection methods.

5. Some reasoning around the Ct threshold of 38 would be helpful to include.

The Ct value of 38 is used because it is recommended by the CDC RT-PCR protocol (reference 26) and is the standard operating procedure used at the Wisconsin State Laboratory of Hygiene. Ct values above 38 are weak positive values indicative of minimal amounts of nucleic acid, which could represent environmental contamination. I could not find any literature to support this outside of manuals/protocols for RT-PCR testing.

6. Paragraph starting at line 125, consider including that differences are statistically significant where appropriate

I added the word “significantly” throughout the results section to identify all statistically significant differences. I also added confidence intervals that were not included in Fig 4 or Fig 5 on lines 167-171.

7. Note: I am concerned about the data sharing. A broad response was given that all data from the Oregon study will be shared, as well as a comment that "some restrictions may apply". It would be helpful to clarify what the restrictions would be and why. Also, to clarify how long after study completion the data will be made available.

The ORCHARDS study and the Household Substudy are ongoing and will continue through at least the 2020/2021 influenza season. All data from this extensive study will be shared upon study completion. For this resubmission, we are making available all data that were used in the analysis, following removal of any personal identifiers. The data have been included in the Supplemental Information (S4 File) as a spreadsheet with all record ID numbers, ages, collection dates, testing dates, RT-PCR results, and human RP Ct values for each participant for whom collection was completed.

.

Reviewer #2:

1. How were confidence intervals calculated? Bootstrap estimates? Were CIs the only statistical means by which differences were tested?

We determined that we could either use 95% confidence intervals for each of the variables or use a t test for the mean Ct value and a X2 test for adequacy. We performed these tests in addition to the confidence intervals and all yielded the same results. We ended up choosing the 95% confidence intervals for the paper because it provides the reader a range of possible values for the mean RP Ct value and the % Adequacy and can be graphed for easy comparison between groups. I also added some description of which confidence interval methods were used on lines 131-135 of the Methods section.

2. I think some further explanation of the sampling would be helpful or even the study population in general. How many household contacts were swabbed etc? For example: Based on the reported numbers in the results, I assume that there were 376 ARI cases reported in school aged children who were swabbed and then enrolled in the follow up study, it then appears that 712 household members collected two follow up swabs, presumably this includes some households (or individual members) who were only able to collect one specimen. Additionally, the original 376 school aged children are part of the group with self-collected specimens. The same questions would apply to year two. Further there could be clustering of specimen quality by household. So it would be good to have a table or flow chart with the number of school-aged ARI cases that triggered collection, the number of households and household contacts, and the number of complete sets of specimens collected.

To address these points, we added a flowchart that shows the total number of screenings, home visits, households, and household participants (Fig 3). It also shows the number of specimens that were untested because of either an incomplete/incorrect collection or a leaking VTM. 

I also added more data to the results section that will help clarify any of these points on lines 144-154. During these edits, I noticed a discrepancy in the written results on line 178 (there were 129 inadequate samples, not 131) that resulted in a 0.1% difference for the total adequacy of participant-collected specimens. This was correctly changed to 96.4% on line 170 and further mention of that value was changed to match it in the Abstract (line 27) and in the Discussion (lines 242 and 306). 

As for the clustering of households, this is addressed in the limitations on lines 315-317. Some households were repeat participants and some were not, so it is possible that the specimen quality is clustered by household. This was not statistically adjusted for in the analysis.

3. I would separate figures into Ct values and adequacy rather than including both on the same plot. Perhaps make separate panels for each comparison.

We agree that separating the Ct values and adequacy would make the graphs easier to read. The two graphs were separated into four graphs (Fig 4-7).

4. I also was confused by place Year 1 and Year 2 next to each other in the Figure 1 plot, when the comparison of interest is between.

We agree with this change and edited Fig 4-7 to match it.

We thank you again for the opportunity to revise and provide changes and clarifications on our manuscript. Should you have any additional comment or question, please do not hesitate to contact me.

Kind Regards,

Mitchell T. Arnold, MPH

---

## [Decision Letter · Decision Letter 1]

28 Aug 2020

Comparison of participant-collected nasal and staff-collected oropharyngeal specimens for human ribonuclease P detection with RT-PCR during a community-based study

PONE-D-20-10148R1

Dear Dr. Arnold,

We’re pleased to inform you that your manuscript has been judged scientifically suitable for publication and will be formally accepted for publication once it meets all outstanding technical requirements.

Kind regards,

Jeffrey Shaman

Academic Editor

PLOS ONE

Additional Editor Comments (optional):

Reviewers' comments:

Reviewer's Responses to Questions

**Comments to the Author**

1. If the authors have adequately addressed your comments raised in a previous round of review and you feel that this manuscript is now acceptable for publication, you may indicate that here to bypass the “Comments to the Author” section, enter your conflict of interest statement in the “Confidential to Editor” section, and submit your "Accept" recommendation.

Reviewer #2: All comments have been addressed

2. Is the manuscript technically sound, and do the data support the conclusions?

Reviewer #2: Yes

3. Has the statistical analysis been performed appropriately and rigorously? 

Reviewer #2: Yes

4. Have the authors made all data underlying the findings in their manuscript fully available?

Reviewer #2: Yes

5. Is the manuscript presented in an intelligible fashion and written in standard English?

Reviewer #2: Yes

6. Review Comments to the Author

Reviewer #2: The authors have responded to all critiques from both of the reviewers in an appropriate manner. No further changes are needed.

7. PLOS authors have the option to publish the peer review history of their article (what does this mean?). If published, this will include your full peer review and any attached files.

Reviewer #2: No

---

## [Editor Report · Acceptance letter]

11 Sep 2020

PONE-D-20-10148R1 

Comparison of participant-collected nasal and staff-collected oropharyngeal specimens for human ribonuclease P detection with RT-PCR during a community-based study 

Dear Dr. Arnold:

I'm pleased to inform you that your manuscript has been deemed suitable for publication in PLOS ONE. Congratulations! Your manuscript is now with our production department. 

Kind regards, 

on behalf of

Prof. Jeffrey Shaman 

Academic Editor

PLOS ONE